# Soybean-Based Polyol as a Substitute of Fossil-Based Polyol on the Synthesis of Thermoplastic Polyurethanes: The Effect of Its Content on Morphological and Physicochemical Properties

**DOI:** 10.3390/polym15194010

**Published:** 2023-10-06

**Authors:** Juliano R. Ernzen, José A. Covas, Angel Marcos-Fernández, Rudinei Fiorio, Otávio Bianchi

**Affiliations:** 1Mantoflex Poliuretanos, Caxias do Sul 95045175, Brazil; julianoernzen@mantoflex.ind.br; 2PGMAT, Universidade de Caxias do Sul (UCS), Caxias do Sul 95070560, Brazil; 3Department of Polymer Engineering, University of Minho, 4800-058 Guimarães, Portugal; jcovas@dep.uminho.pt; 4Elastomers Group, Institute of Polymer Science and Technology (ICTP-CSIC), 28006 Madrid, Spain; 5Interdisciplinary Platform for “Sustainable Plastics towards a Circular Economy” (SUSPLAST-CSIC), 28006 Madrid, Spain; 6Department of Circular Chemical Engineering, Maastricht University, 6200 MD Geleen, The Netherlands; r.fiorio@maastrichtuniversity.nl; 7Department of Materials Engineering, Universidade Federal do Rio Grande do Sul (UFRGS), Porto Alegre 90040040, Brazil

**Keywords:** thermoplastic polyurethane, soybean polyol, reactive extrusion, structure–properties relationship

## Abstract

Thermoplastic polyurethanes (TPUs) are remarkably versatile polymers due to the wide range of raw materials available for their synthesis, resulting in physicochemical characteristics that can be tailored according to the specific requirements of their final applications. In this study, a renewable bio-based polyol obtained from soybean oil is used for the synthesis of TPU via reactive extrusion, and the influence of the bio-based polyol on the multi-phase structure and properties of the TPU is studied. As raw materials, 4,4′-diphenylmethane (MDI), 1,4-butanediol, a fossil-based polyester polyol, and a bio-based polyol are used. The fossil-based to soybean-based polyol ratios studied are 100/0, 99/1, 95/5, 90/10, 80/20, and 50/50% by weight, respectively. The TPUs were characterized by size exclusion chromatography (SEC), gel content analysis, Fourier transform infrared spectroscopy (FTIR), differential scanning calorimetry (DSC), small-angle X-ray scattering (SAXS), dynamic mechanical analysis (DMA), and contact angle measurements. The results reveal that incorporating the renewable polyol enhances the compatibility between the rigid and flexible segments of the TPU. However, due to its high functionality, the addition of soybean-based polyol can promote cross-linking. This phenomenon reduces the density of hydrogen bonds within the material, also reducing polarity and restricting macromolecular mobility, as corroborated by higher glass transition temperature (T_g_) values. Remarkably, the addition of small amounts of the bio-based polyol (up to 5 wt.% of the total polyol content) results in high-molecular-weight TPUs with lower polarity, combined with suitable processability and mechanical properties, thus broadening the range of applications and improving their sustainability.

## 1. Introduction

Polyurethanes (PUs) are found in many everyday use articles, such as cushions, pillows, and mattresses. Particularly, thermoplastic polyurethanes (TPUs) are one of the most valuable thermoplastic elastomers (TPEs), finding extensive applications in distinct fields such as biomedicine, automotives, civil construction, electronics, and paints. Furthermore, TPUs exhibit notable mechanical properties and excellent biocompatibility [1,2]. TPUs represent a special class of polyurethanes containing two different macromolecular segments: a soft segment (SS) derived from a macrodiol, and a hard segment (HS) composed of a diisocyanate and a low-molecular-weight diol [1,2]. The thermodynamic incompatibility between SS and HS leads to the formation of two microphases, a soft phase (SP) and a hard phase (HP), each one with distinct glass transition temperatures (T_g_) and/or melting points. The microphase separation is influenced by factors such as polarity, solubility parameters, and hydrogen bonding formation, and is pivotal in determining the morphology and physicochemical properties of TPUs [3,4,5].

The soft phase plays a key role in the elastomeric properties of TPUs. This phase influences final properties such as thermal stability [6,7,8], biodegradation [9], hydrolysis resistance [10] and mechanical properties. For instance, the amount of hydrogen bonds in the soft phase directly affects the elastic modulus of TPUs [11,12].

The blending of conventional fossil-based polyols with bio-based polyols from renewable sources has gained substantial attention, leading to the development of more sustainable TPUs. This simple approach potentially reduces the environmental impact of polyurethane production, while providing unique physicochemical properties [1]. Nevertheless, the impact of incorporating bio-based polyols on the structural and property attributes of TPUs and its influence on final product characteristics remain relatively underexplored.

Previous studies have observed that the use of bio-based polyols typically modifies the polarity of the soft phase of TPUs. An increase in the polarity of the flexible phase can lead to a substantial improvement in the hydrolysis resistance of the material [13]. Nevertheless, the presence of secondary hydroxyl groups commonly found in bio-based polyols usually impacts their reactivity and the TPUs molecular configuration [14]. Luong and coworkers [15] prepared a series of TPUs in which the flexible phase consisted of poly(tetrahydrofuran) polyol, poly(dimethylsiloxane), and up to 14.1 wt.% castor oil. These authors observed that the TPUs maintain their thermoplastic characteristics and show large elongations upon breaking (up to 1200%). They also noticed the formation of branched macromolecular structures, and advocate that TPUs containing castor oil have potential biocompatibility.

Some limitations of polyols based on castor oil, or polyols obtained by epoxidation of soybean oil, for the synthesis of TPUs are related to their high functionality (f > 2) and the possible formation of gel structures [16], which can make it difficult to recycle the material using conventional techniques such as extrusion and injection due to the formation of crosslinks [17]. Although there are other alternatives for the development of bio-based polyols, such as using dimerized acids from vegetable oils [18,19,20], the production of these materials on a large scale is still a challenge, as there is still an industry restriction on their use due to reaction conditions already well established in polymerization plants, and other factors such as polyol composition [14,21].

Hence, gaining insight into the impacts of blending bio-based polyols with fossil-derived counterparts on the morphological and physicochemical attributes of TPUs is essential for advancing the adoption of renewable polyols over unsustainable alternatives. To address this, we propose employing polyol blends in the synthesis of thermoplastic polyurethanes via reactive extrusion.

In the current study, we highlight the effect of soybean-based polyol on the physicochemical and morphological characteristics of TPUs obtained via reactive extrusion. TPUs were investigated via size exclusion chromatography (SEC), gel content, Fourier transform infrared spectroscopy (FTIR), differential scanning calorimetry (DSC), small-angle X-ray scattering (SAXS), surface energy, and dynamical mechanical analysis (DMA). Finally, the mechanical properties of two extruded articles (pneumatic tubes) were evaluated. Tube geometry was selected, as it requires a polymer with high molar mass and flow stability during molding.

## 2. Materials and Methods

The isocyanate employed in the synthesis of the TPUs is 4,4′-diphenylmethane diisocyanate (MDI) (ISONATE 125M; Dow Chemical, São Paulo, Brazil). The MDI has a 33.5 wt.% of isocyanate (NCO) groups and a density (ρ) of 1.23 g.cm^−3^. The chain extender is 1,4-butanediol (BDO) with a density of 1.02 g.cm^−3^, also supplied by Dow Chemical, São Paulo, Brazil. The conventional, fossil-based polyester polyol (YA7220, Coating P. Materials, Taichung, Taiwan) shows a molecular weight of 2000 g.mol^−1^, a hydroxyl number of 55–60 mgKOH.g^−1^, and a density of 1.15 g.cm^−3^. Additionally, a bio-based polyol is used, which was obtained via epoxidation of soybean oil, exhibiting a hydroxyl number of 90 mgKOH.g^−1^, a density of 0.97 g.cm^−3^, a molecular weight of 2000 g.mol^−1^, and a functionality of 3.2. This polyol was synthesized according to the literature [22].

### 2.1. Reactive Extrusion and Injection Molding

Different polyol compositions (100/0, 99/1, 95/5, 90/10, 80/20, and 50/50% by weight of the fossil-based and the bio-based polyols, respectively) were prepared at 80 °C and stirred for 15 min. Next, pre-polymers were prepared in a 5 L stainless steel vessel by incorporating MDI into the polyols. The resulting mixture was maintained at 80 °C under mixing during the reactive extrusion step. An inert atmosphere (N_2_) was employed to prevent degradation and side reactions. A chain extender was added separately at the beginning of the extrusion screw. In the reactive extrusion process, all TPUs were synthesized with an NCO/OH ratio of 1.05/1 and approximately 50 wt.% of hard segments (HS). The production rate of TPUs was approximately 2 kg.h^−1^. The extruder used was an LTE 16–48 co-rotating twin-screw extruder (LabTech Engineering Company Ltd., Praksa, Thailand), presenting a length-to-diameter ratio (L/D) of 48. The processing temperatures ranged from 160–220 °C, and a screw speed of 250 rpm was utilized. The screw profile, optimized for the synthesis of TPU was based on previous studies conducted by our research group [12]. Under these processing conditions, the average residence time was approximately 300 s.

The extruded polymers were cooled in water at 25 °C and pelletized for further characterization. Additionally, the materials were injection-molded (homemade) at 250 °C and 90 bar to obtain specimens for the subsequent SAXS, DMA, and physical–mechanical tests (ISO 527-2/1BA/500).

### 2.2. Gel Content and Size Exclusion Chromatography (SEC)

The gel content of the TPUs was determined following the ASTM D2765 standard [23]. A 120-mesh wire cage containing approximately 0.3 g of polymer was placed in a round-bottomed flask to estimate the gel content. The flask was filled with boiling dimethylformamide (DMF) and left for 8 h for solvent extraction. Following the extraction process, all samples were subsequently dried at 60 °C for 48 h to ensure complete removal of any residual solvent and moisture.

The molecular weight of the TPUs was determined using size exclusion chromatography (SEC) on a Perkin-Elmer series 200 chromatograph (PerkinElmer, Waltham, MA, USA) equipped with a refractive index detector. The eluent employed was dimethylformamide containing 1% of BrLi. The characterization process involved a sample concentration of 10 mg.mL^−1^, a flow rate of 1 mL.min^−1^, an injected volume of 10 μL, and a column temperature of 35 °C. Polystyrene standards were employed for the calibration curve.

### 2.3. Fourier Transform Infrared Spectroscopy (FTIR)

Fourier transform infrared spectroscopy (FTIR) was utilized to assess the chemical structure of the TPUs and investigate the impact of the renewable polyol on phase separation. FTIR analysis was performed using a Perkin-Elmer Spectrum 400 spectrometer in the attenuated total reflectance (ATR) mode, with a diamond crystal set at 45°. The samples were scanned 32 times within the range 4000–450 cm^−1^ at a resolution of 2 cm^−1^. Furthermore, the degree of hydrogen bonding and the level of hard segments dispersed in the soft phase were determined through mathematical deconvolution, as described in [24].

The FTIR absorption bands between 1500–1800 cm^−1^, associated with the carbonyl group (C=O), are sensitive to the amount and type of hydrogen bonding in polyurethane materials. To elucidate the relative quantities of the different types of hydrogen bonding within the samples, we focused first on evaluating the interactions in the polyols (PM). The absorption band at approximately 1745 cm^−1^ (band 1 PM) corresponds to free carbonyl groups. The absorption band at around 1730 cm^−1^ (band 2 PM) is linked to carbonyl groups physically bonded through dipole–dipole interactions, while the absorption band at approximately 1700 cm^−1^ (band 3 PM) relates to carbonyl groups physically attached via hydrogen bonds between the terminal –OH groups of the macrodiol and the C=O of ester groups [17]. The absorption peaks were resolved by deconvolution, and the results are provided in Appendix A.

Afterwards, the effect of the renewable polyol content on the phase separation was assessed by considering the contributions of six bands in the range of 1500–1800 cm^−1^ [17,24,25]. Analyzing the absorption bands in this region gives an estimation of the degree of phase separation within the TPUs. Previous studies have employed these bands to determine the relative amount of hydrogen bonding [4,17,24]. Mathematical adjustments were made for each spectrum, assuming Gaussian functions (see Appendix A). The six bands were described as follows: (I) at 1675 cm^−1^, H-bonded urethane carbonyl groups (ordered phase–hard segments); (II) at 1699 cm^−1^, H-bonded urethane carbonyl groups (disordered phase-hard segments); (III) at 1710 cm^−1^, H-bonded carbonyl groups (soft–hard segment); (IV) at 1722 cm^−1^, free carbonyl from urethane groups (hard segment); (V) at 1732 cm^−1^, carbonyl-carbonyl interactions (soft segment); and (VI) at 1745 cm^−1^, free carbonyl (soft segment). The weight fraction of the hydrogen-bonded urethane groups (X_b_), weight fraction of the hard segment dispersed in the soft segment (W_h_), mixed-phase weight fraction (MP), soft-phase weight fraction (SP), and hard-phase weight fraction (HP) were determined using Equations (1)–(5), respectively [17]:(1)$$X_{b}=\frac{A_{b}}{k’A_{f}+A_{b}}=\frac{A_{1675}+A_{1699}}{{k^{’}A}_{1722}+A_{1675}+A_{1699}}$$
(2)$$W_{h}=\frac{\left(1-X_{b}\right)f}{\left[\left(1-X_{b}\right)f+\left(1-f\right)\right]}$$
(3)$$MP={fW}_{h}$$
(4)SP=MP+(1−f)
(5)HP=1−SP
where $A_{b}$ is the area associated with H-bonds between two urethane groups (bands I and II), $A_{f}$ is the absorbance of free carbonyl from non H-bonded urethane groups (band IV), $k^{’}$ is the extinction coefficient (a constant equal to 1.2) [24], and f is the weight fraction of hard segments in the polymer, determined from the initial molar ratios.

### 2.4. Differential Scanning Calorimetry (DSC)

The thermal transitions of the polyols and TPUs were determined using differential scanning calorimetry (DSC) with a Mettler Toledo DSC 822e instrument (Mettler Toledo, Columbus, OH, USA), operating under a nitrogen atmosphere at a flow rate of 50 mL/min. Sample sizes of 9–10 mg were utilized. The melting temperature and enthalpy values were calibrated using indium, tin, bismuth, and zinc standards. The DSC analyses were performed over a temperature range of −80 to 240 °C, with a heating rate of 20 °C/min. The glass transition temperature (T_g_) was determined at the onset point, while the variation of heat capacity (ΔC_p_) was calculated at the midpoint of the T_g_ range. To ascertain the specific transition temperatures of the rigid phase, a rigid phase model compound consisting of MDI and BDO (NCO/OH = 1.0/1.0) was prepared at 80 °C using mechanical stirring for 1 h in nitrogen atmosphere.

### 2.5. Small Angle X-ray Scattering (SAXS)

Small-angle X-ray scattering (SAXS) experiments were conducted in the SAXS1 beamline of the Brazilian Synchrotron Light Laboratory (LNLS). The experimental setup involved monitoring with a photomultiplier and detection with a Pilatus 300K detector (300 k, 84 mm × 107 mm; Rigaku Corporation, Tokyo, Japan), positioned at a distance of 836 mm from the sample. The scattering wave vectors (q) ranged from 0.13 to 2.5 nm^−1^, and the incident X-ray beam had a wavelength (λ) of 0.155 nm. The samples used in the experiments were 3 mm in diameter and 1 mm thick. The diffraction angle was calibrated using a silver behenate (AgBeH) standard. The measurements were performed isothermally, ranging from 23 to 220 °C. In separate measurements, the background and parasitic scattering were determined using an empty sample holder, and these signals were subtracted from the scattering signals of the samples.

Considering that all TPUs have about 50 wt.% of rigid phase, one can assume a co-continuous morphology of their separated microphases. Thus, the high q scattering peak was analyzed using the Teubner–Strey model. This model, based on the Landau–Ginzburg free-energy theory, describes a two-component structure when particle shape is not well defined, and has been successfully applied to a range of two-phase non-particulate systems [26,27]. The Teubner–Strey model predicts a scattering intensity as follows [28]:(6)$$I\left(q\right)=A\frac{1}{a_{2}+c_{1}q^{2}+c_{2}q^{4}}$$

In the context of an order parameter expansion of the free energy density, the coefficients a_2_, c_1_, and c_2_ play crucial roles. Additionally, A is a normalization parameter that correlates with the volume fraction of a specific phase and its scattering contrast. Equation (6) defines the scattering intensity, which is linked to the correlation function of the Teubner–Strey model through a direct Fourier transform. Additionally, it serves as a normalization parameter that correlates with the volume fraction of a specific phase and its scattering contrast. The correlation function can be expressed as follows [26]:(7)$$\gamma \left(r\right)=\frac{d}{2\pi r}exp\left(-\frac{r}{\xi }\right)sin\left(\frac{2\pi r}{d}\right)$$
where r is the real-space distance, *ξ* is the correlation length, and d is related to the d-spacing average repeating distance.

### 2.6. Surface Energy

The surface free energies of the TPU were determined using the Owens–Wendt method, which is based on contact angle measurements conducted with standard liquids [29,30]. The wettability behavior of the solid specimens provides a better understanding of the adhesion phenomenon. The contact angle measurements were carried out using an SEOVR Phoenix100 (Suwon, Republic of Korea) instrument. Six liquids were employed at 23 ± 2 °C: distilled water (γ_L_^P^ 51.0 mJ/m^2^, γ_L_^D^ 21.8 mJ/m^2^, and γ_L_ 72.8 mJ/m^2^), glycerin (γ_L_^P^ 29.7 mJ/m^2^, γ_L_^D^ 33.6 mJ/m^2^, and γ_L_ 63.4 mJ/m^2^), ethylene glycol (γ_L_^P^ 19.0 mJ/m^2^, γ_L_^D^ 29.0 mJ/m^2^, and γ_L_ 48.8 mJ/m^2^), dimethylformamide (γ_L_^P^ 37.3 mJ/m^2^, γ_L_^D^ 32.4 mJ/m^2^, and γ_L_ 4.9 mJ/m^2^), dimethyl sulfoxide (γ_L_^P^ 8.0 mJ/m^2^, γ_L_^D^ 36.0 mJ/m^2^, and γ_L_ 44.0 mJ/m^2^), and hexadecane (γ_L_^P^ 0.0 mJ/m^2^, γ_L_^D^ 27.6 mJ/m^2^, and γ_L_ 27.6 mJ/m^2^), where γ_L_^P^, γ_L_^D^ and γ_L_ represent the polar component, the dispersive component, and the surface free energy of the liquids, respectively. The sessile drop method was adopted using 2 μL drops. The contact angle was measured at least ten times at different sites on the surface to consider the average value. From the collected data, the surface energy and polar and dispersive components of the polymers were estimated as described elsewhere [30].

### 2.7. Dynamical Mechanical Analysis (DMA)

Dynamic mechanical experiments were conducted using a DMA Q800 (TA Instruments, New Castle, DE, USA) employing a dual cantilever geometry. The experiments were carried out within the linear viscoelastic region, at a small amplitude of 15 μm, over a temperature range of −70 to 180 °C. A fixed heating rate of 2 °C/min was used, at frequencies of 1, 3, 10, 30, and 100 Hz. Rectangular test specimens of ca. 35 mm × 10 mm × 3.2 mm were obtained from injection-molded specimens.

### 2.8. Physical–Mechanical Properties and Application in Tubing

After characterizing the materials, we proceeded to manufacturing, via plasticating extrusion, two tubes with dimensions of 8 mm × 1.25 mm, and 6 mm × 1.00 mm (outer diameter × wall thickness, respectively). They were produced based on two TPUs, one containing only fossil-based polyol, the other consisting of 5 wt.% soybean and 95 wt.% fossil-based polyols. These materials were synthesized, pelletized and dried, followed by extrusion in a single-screw extruder (L/D 35 30 rpm, 180–220 °C, Miotto, São Bernardo do Campo, Brazil). During the processing of the tubes, the barrel temperatures were set in the range 180–210 °C, and a screw speed of 30 rpm was used. About 1 wt.% of blue pigment was added to this TPU for the manufacture of tubes to mimic tubes used industrially.

To assess the performance of the two TPUs, physical–mechanical characterizations were carried out to evaluate their properties and performance using injection-molded samples. Hardness and tensile properties were evaluated. To determine the hardness of the samples, a durometer Woltest SD 100 (Woltest Comercial Ltda, São Paolo, Brazil) was used, in line with the ASTM D2240-15 standard [31]. Tensile tests followed the ISO 527-2/1BA/500 standard [32], using an Instron machine (EMIC DL 30000; Instron, Norwood, MA, USA) equipped with an extensometer at a crosshead speed of 500 mm/min. Additionally, the burst pressures of the produced tubes were evaluated using compressed air. This comprehensive approach allows one to assess the suitability and quality of the produced tubes, providing valuable insights for potential applications according to final requirements.

## 3. Results and Discussion

### 3.1. Gel Content and Molecular Weight

The gel content analysis indicated that the new TPUs exhibited gel formation when the soybean-based polyol content reached 10 wt.%. Specifically, 7 wt.% and 10 wt.% of gel content were observed for the samples containing 20 wt.% and 50 wt.% bio-based polyol, respectively. This result is attributed to the high functionality of the polyol (f > 2) [1], which induces the formation of cross-links. Cross-links can pose challenges during processing as well as during solubilizing TPUs. The TPUs synthesized with the incorporation of soy polyol exhibited a subtle yellowish hue, primarily attributed to the inherent color of the monomer, as depicted in Appendix A.

Table 1 shows the size exclusion chromatography (SEC) results. One can observe that the number average molecular weight (M_n_) and weight average molecular weight (M_w_) values are high, indicating successful syntheses. Additionally, the molecular weight distribution (dispersity, *Đ*) values align with those typically observed for TPUs synthesized via reactive extrusion [12]. Notably, M_n_, M_w_, and *Đ* values increased with the addition of up to 5 wt.% bio-based polyol, suggesting the possible formation of branched molecules [15]. However, due to cross-linking reactions and the formation of gels, the molar mass of the extracted fraction decreased for samples containing 10 wt.% or more of the bio-based polyol. Therefore, samples containing 1–5 wt.% renewable polyol can present advantages, as the higher polydispersity enhances the flow stability of TPU during processing [2].

### 3.2. Fourier Transform Infrared Spectroscopy (FTIR)

From the FTIR spectra, the weight fraction of hydrogen-bonded urethane groups (X_b_), weight fraction of hard segments dispersed in the soft segments (W_h_), weight fraction of the mixed phase (MP), weight fraction of the soft phase (SP), and weight fraction of the hard phase (HP) were determined and are presented in Table 2. For all TPUs, a high fraction of hydrogen-bonded urethane groups was noted (Appendix A). However, a more pronounced reduction was observed for the samples containing 20 and 50 wt.% of soybean-based polyol. This fact is related to the lower polarity of the polyol due to the olefinic groups (–CH_2_–). Regarding the W_h_ fraction, no significant difference was noticed among the samples, showing that virtually the same amount of rigid phase is dispersed in the flexible phase. The values of MP (0.13–0.14), SP (0.67–0.68), and HP (0.32–0.33) remained practically constant. The results shown in Table 2 agree with previous data reported in the literature for TPUs, showing well-separated hard and soft phases [33].

The decrease in the value of X_b_ with the increase in the content of soybean-based polyol is directly linked to the extent of phase separation in polyurethanes [17]. Specifically, a higher relative number of hydrogen bonds results in a more cohesive rigid domain. However, several factors can reduce X_b_ and subsequently affect phase separation. When the solubility parameters of the rigid and flexible phases are similar, phase separation is usually reduced. Additionally, the formation of cross-links can hinder the mobility of the phases, limiting their separation by restricting macromolecular movements [25,34].

### 3.3. Differential Scanning Calorimetry (DSC)

The glass transition temperatures observed for the polyols (T_gss_) were −24 °C for the fossil-based polyol and −80 °C for the soybean-based polyol [22]. The DSC results of the TPUs are shown in Figure 1a. Two thermal events are clearly observed: the glass transition temperature of the soft phase (T_gSP_) and the melting temperature (T_m_). The former showed direct dependence on the polyol composition. As for T_m_, the samples consistently exhibited two melting peaks, occurring at 206 and 213 °C, irrespective of their composition. However, in the case of a sample with 50 wt.% bio-based polyol, an additional melting peak was observed at 181 °C. This peak can be ascribed to a localized restructuring of the hard segments within the rigid microdomains, leading to the generation of smaller crystals. This phenomenon arises from the formation of cross-links, which in turn restricts the mobility of macromolecules [34]. According to previous studies, multiple melting peaks can be linked to reorganization processes that occur during heating, involving repeated melting and recrystallization steps [35]. The underlying mechanisms responsible for this phenomenon have been attributed to various factors such as local restructuring of hard segments within the hard microdomains [36,37], the disruption of hard-segment structures mainly composed of MDI-BDO-MDI segments [38,39], a perceived glass-transition process occurring within the hard microdomains, the formation of short-range ordered structures facilitated by hard-segment sequences at the annealing temperature, and the enthalpy relaxation of the amorphous hard segment [40].

Upon the addition of soybean-based polyol, a noticeable increase in the range wherein T_m_ and T_g_ occur is observed, which affects the thermal behavior of the TPUs, as depicted in the Appendix A. The formation of crosslinks imposes molecular constraints, disturbing the crystallization of the rigid segment during cooling. However, crystallization was observed during the second heating cycle. Examining the values of enthalpy of fusion (ΔH_m_), as depicted in Figure 1b, the TPU initially exhibited an ΔH_m_ of ca. 14 J/g. Notably, when soybean polyol was added up to 5 wt.%, an increase in the enthalpy of fusion is observed (~20 J/g). This increase could potentially be attributed to the formation of branched structures, which contribute to an initial increase in free volume, thereby facilitating the crystallization process. Nevertheless, with a further increase in bio-based polyol, as gel structures are formed, there is a gradual restriction of the macromolecular mobility, leading to a subsequent reduction in ΔH_m_. Thus, forming gel structures ultimately results in a decrease in the enthalpy of fusion.

Table 3 presents the estimated T_gSP_, ΔC_p_, and phase separation data, following Camberlin and Pascault’s criteria, comparing heat capacity variation between TPU and pure polyols [41]. Regarding the glass transition temperature of the soft phase (T_gSP_), there is an increase in mobility within the soft phase for samples containing up to 5 wt.% of soybean-based polyol, leading to a decrease in the glass transition values. However, owing to the high polyfunctionality of soybean polyol, the T_gSP_ values increase starting from the sample containing 10 wt.% of bio-based polyol. This rise in T_gSP_ may be attributed to two factors: (i) the formation of crosslinks, resulting in segmental restriction; and (ii) the reduction in the difference in solubility parameters between the rigid and flexible phases, as reported in the literature [12,16,17].

It is evident that the quantity of soybean polyol strongly influences the extent of phase separation. The degree of phase separation (DPS) shows a linear decrease with the increase in polyol ($DPS=-0.71\;*\;wt.\%\;soybean\;polyol\;+77.6,\;R^{2}>0.964$). This correlation suggests that the reduction in phase separation is primarily driven by the higher crosslinking density formed, rather than solely by the molecular mobility restriction [25,34]. Considering the solubility parameters (δ) of the non-renewable polyol (approximately 19.9 MPa^1/2^ [12]), the soybean polyol (ca. 24.8 MPa^1/2^ [17]), and of the rigid phase consisting only of MDI-BDO (25.4 MPa^1/2^ [17]), one can infer that the bio-based polyol exhibits a higher affinity—and thus better miscibility—with the rigid phase than the fossil-based polyol. However, the soybean polyol shows a lower T_gSS_ due to its olefinic segments. This observation suggests that the formation of crosslinks affects the phase separation phenomenon to a greater extent than the chemical affinity of the different segments.

### 3.4. Small Angle X-ray Scattering (SAXS)

From the SAXS results, morphological models were employed to characterize the scattering intensities resulting from microphase separation. The morphologies of phase-separated segmented polyurethanes depend on the concentrations of both hard and soft segments [42]. When the concentration of hard segments is below 30%, isolated hard domains are typically formed and dispersed within a matrix consisting of soft segments (the soft phase). On the other hand, when the concentration ratio between hard and soft segments is 1:1, the free energy constraints drive the formation of a co-continuous morphology [27,43,44].

For the samples containing 0 up to 10 wt.% of soybean polyol, a distinct correlation peak (q close to 0.6 nm^−1^) was observed (Appendix A), associated with phase-separated domains; this is depicted in Figure 2a for the reference TPU sample containing only fossil-based polyol. Furthermore, within the range of 0.12 < q < 0.25 nm^−1^ and at temperatures ranging from 30 to 120 °C, a reduction in the exponent value from q^−2.89^ to q^−0.69^ was observed. However, when the temperature surpassed 170 °C, this power-law behavior shifted to a range of 0.5 < q < 1 nm^−1^. This modification is directly linked to the melting of the crystalline domains and the loss of the rigid phase ordering in the TPUs. At 220 °C, the sample was completely molten, and the scattering curve profile exhibits characteristics of amorphous polymers, lacking long-range order [10].

Concerning the addition of soybean polyol, it can be observed that the correlation peak at 0.6 nm^−1^ tends to broaden and virtually disappears for the sample containing 50 wt.% of soybean polyol at 30 °C (Figure 2b). This behavior is attributed to the cross-linking and segmental restriction of this sample, which hinder crystallization. However, in temperatures above 100 °C, the macromolecular segments’ mobility increases, and the rigid segments may reorganize. At 30 °C, the exponent takes a value of q^−3.82^, which shifts to q^−4^ at 100 °C and q^−3.71^ at 220 °C. In general, the values obtained with the addition of soybean polyol are lower than those for the reference TPU, suggesting that the interfaces/surfaces are smoothed. Alternatively, the domains may contain diffuse interfaces in cases wherein the slope values are close to −4 [45].

The long period (L_p_) of the samples was estimated as a function of temperature by plotting I(q).q^2^ vs. q (Lorentz correction), and the results are depicted in Figure 3. As the temperature increased, L_p_ also increased, indicating enhanced mobility and larger separation between the scattering centers. Nevertheless, with the addition of 50 wt.% bio-based polyol, the distances between the scattering centers were reduced. This reduction is attributed to the restriction of segmental mobility caused by the formation of cross-links, reducing the separation between domains and occasionally raising the order–disorder temperature of the blocks. Other samples show a similar trend, with differentiation being evident.

Since the samples contain approximately an equivalent weight ratio of rigid and flexible phases (~50% each), their morphology is considered co-continuous. The Teubner-Strey model is applicable to non-particulate two-phase systems, including microemulsions with ill-defined particle shapes [26,27,46], and it was used to analyze the high-q scattering peak in the range of 30–200 °C for the samples containing up to 20 wt.% of soybean polyol. However, for the sample with 50 wt.%, this model fitted the experimental results only at temperatures above 100 °C.

The Teubner–Strey model adequately describes the high-q scattering peak, as evidenced by the high correlation coefficient (>0.99) for all samples, as shown by the red lines in Figure 2a,b. The model provides correlation lengths (ξ) and mean repetition distances (d), illustrated in Figure 4a,b. The length scale, d, represents a quasi-periodic repetition distance between scattering center regions within the flexible phase. In contrast, the correlation length corresponds to a characteristic length for positional correlation, or the length scale of the scattering objects [26].

Both correlation length (ξ) and mean repetition distances (d) increase with temperature, a typical indication of an increase in the system mobility. Notably, ξ increases for the sample with 1 wt.% soybean polyol, since even small concentrations of this polyol can lead to chain branching and consequently enhance the molecular mobility. In contrast, when crosslinks are formed, inducing the formation of gels, segmental restriction occurs, and the size of the organized phase is reduced. Regarding d, the trend is similar to the L_p_ data, further supporting the influence of molecular mobility on the TPUs structural characteristics.

### 3.5. Dynamical Mechanical Analysis (DMA)

From the dynamic mechanical analysis (DMA), the storage modulus (E′) as a function of temperature for the TPUs, with data collected at 1 Hz, is shown in Figure 5a. The Appendix A provide additional results obtained in different conditions. Figure 5a indicates an elastic pseudo-solid behavior for all samples up to 170 °C, below the onset of the melting temperature of the rigid phase, as observed in the DSC and SAXS results. Overall, all curves exhibit distinct regions, including the glassy plateau, the glass transition region, the elastic plateau, and the onset of the flow region.

Evidently, the glass transition temperature exhibits a tendency to decline as soybean polyol content increases, a trend observed until the point of cross-linking initiation. This phenomenon is attributed to the influence of soybean polyol, which impacts the glass transition temperature of the olefinic portion, and consequently leads to this effect [22]. The presence of cross-links strongly affects the glass transition region, which is particularly evident in the temperature range of 0−75 °C. The formation of cross-links restricts cooperative movements, leading to an increase in the glass transition temperature, as previously presented in Table 3. For the reference sample without soybean polyol, the measured T_g_ was 15.5 °C, which increased to 62.3 °C for the sample containing 50 wt.% soybean polyol. A similar trend was observed in the DSC data.

The shift of the elastic plateau region towards higher temperatures is a direct result of the increased soybean polyol content, primarily attributed to the formation of cross-links. These cross-links have the effect of impeding the dynamics of the polymer chains, which subsequently necessitates a higher temperature to achieve the desired elasticity [47,48]. It is worth highlighting that with increasing soybean polyol content (as illustrated in Appendix A), the plateau’s extension becomes narrower. This phenomenon can be attributed to the formation of partial cross-links, originating from regions characterized by unique conformational energy barriers. As the degree of crosslinking intensifies, it gives rise to clusters of crosslinked material, each demonstrating cooperative movements at distinct temperature ranges [48]. Consequently, this diversity in cooperative behavior contributes to a broader plateau. Conversely, when the heterogeneity diminishes, and the population of cooperative movements becomes more uniform, it leads to an extension of the plateau. However, it is essential to note that this particular type of TPU cannot form a three-dimensional network akin to traditional thermosets like epoxy resin, primarily due to disparities in its macromolecular composition [47].

The storage modulus provides insights into the average distance between the crosslink points (M_c_) and the cross-linking density ($\nu_{e}=1/M_{c}$). The theory of rubber elasticity indicates the following correlation between the torsional storage modulus and M_c_ [49]:(8)$$G^{’}=\frac{\rho RT}{M_{c}}$$

To calculate the shear modulus from the storage modulus measured in the DMA under bending conditions, we can utilize the Poisson ratio (ν), i.e., the ratio of transverse strain to axial strain when a material is subjected to applied stress. The relationship between the storage modulus (E′) obtained from the DMA and the shear modulus (G) can be expressed as follows:(9)$$G^{’}=\frac{E’}{2(1+\nu )}$$

To determine the M_c_ value from the storage modulus values in the elastic plateau region, one can assume that
(10)$$M_{c}=\frac{2(1+\nu )\rho RT}{{E’}_{0}}$$

A ν value of 0.4 was assumed for all samples. It was noted that a higher amount of soybean polyol reduced the density from 1.1 g/cm^3^ to 0.97 g/cm^3^. To determine a frequency-independent modulus value, adjustments to the modulus data collected in the elastic plateau were performed as a function of different frequencies using an exponential equation, as illustrated in the Appendix A and Table 4. Initially, there is an increase in the values of E_e′_ (storage modulus) followed by a subsequent reduction, indicating that as the number of crosslinks increases, the TPUs exhibit a more pronounced elastic behavior due to the reduction in organized domains.

Although the values of ν_e_ (crosslinking density) may not hold an actual meaning for TPUs given that the material does not naturally present crosslinks, they provide insights into the effect of soybean polyol on the material. As expected, increasing soybean polyol content leads to an increase in crosslink density, as previously evidenced by the gel content results and the preparation of solutions for the GPC experiments. Consequently, with an increased amount of the cross-linked phase, there is a reduced relative contribution of the rigid phase, which is a crucial component affecting the mechanical behavior. Therefore, there is a reduction in the extent of phase separation, leading to lower values of E_e_′ [34].

Additionally, from the DMA results, the activation energy of segmental mobility (E_a_), which is closely related to the molecular mobility, can be estimated using an Arrhenius relation, expressed as follows [50]:(11)$$Lnf=LnC-\frac{E_{a}}{RT}$$
where f is the frequency, C is a constant, R is the universal gas constant, and T is the temperature corresponding to the peak in the tan δ curve at different frequencies. By plotting Ln f vs. 1/T, a straight line is obtained, as shown in Figure 5b, and E_a_ is obtained from the slope of the linear regressions. Determination coefficients greater than 0.98 were obtained for all samples. The E_a_ values of the samples containing 1 and 5 wt.% soybean polyol are lower than that for the reference TPU without soybean polyol; this can be attributed to branching and increased free volume found in the TPUs with soybean polyol. As expected, as the soybean polyol content rises, leading to increased segmental restriction due to crosslinking, E_a_ gradually increases. Table 4 summarizes the results obtained from the DMA analysis.

### 3.6. Contact Angle Measurements

Figure 6 shows contact angle results for the TPU samples using different solvents, and Table 5 presents the polar (γ_S_^P^) and dispersive (γ_S_^D^) components, as well as the surface energies (γ_L_) of the TPUs. Overall, Figure 6 shows that the contact angle increased with the addition of soybean polyol for practically all solvents, except for hexadecane, which is a non-polar hydrocarbon molecule. It is known that a higher contact angle indicates lower wettability. Therefore, the contact angle results indicate that the addition of soybean polyol reduced the polarity of the TPUs, thus increasing their hydrophobic characteristics. This behavior can be attributed to the molecular structure of the polyol, which presents longer hydrocarbon segments than the fossil-based polyol. This reduction in polarity effect is evident from Table 5, wherein the dispersive and polar components, as well as the surface energy values, are lower for the TPUs containing more than 1 wt.% soybean polyol compared to the reference TPU. Appendix A displays instances of droplets employed for contact angle measurements on TPU samples in water. These examples effectively illustrate the reduction in polarity.

According to Ourique et al. [10], polyurethanes derived from vegetable oils, such as those incorporating soybean polyol, can present higher resistance to hydrolysis due to the olefinic structure found in these polyols. As polyester-based TPUs typically present limited resistance to hydrolysis, incorporating soybean polyol can offer promising alternatives for improving their durability and performance in various applications.

### 3.7. Mechanical Properties

Table 6 outlines the findings from the physical–mechanical evaluation of two materials: the reference TPU (devoid of soybean polyol) and the TPU infused with 5 wt.% of soybean polyol. Both materials boast practically indistinguishable hardness levels. However, the reference TPU outperforms the 5 wt.% soybean polyol-laden variant in stress at 100% strain, which can be attributed to cross-link formation within the TPU. At higher strains (300%), the 5 wt.% soybean polyol-infused sample exhibits an increase in tensile strength, likely due to its higher molecular mass and the formation of cross-linked regions. Considering that the mechanical characteristics of TPUs are significantly dependent on their hydrogen bond density [12,17], it is challenging to draw any inference regarding their effects in this specific context due to their similarity, according to Table 2. Consequently, it seems that the enhancement of mechanical properties is primarily driven by the increase in molar mass and the formation of crosslinks.

Notably, disparities were found between the material’s surface during the extrusion process for tube manufacturing. The inclusion of soybean polyol triggers lump formation due to the polymer’s increased molecular mass, necessitating screw speed and temperature adjustments. Nonetheless, both 8 mm outer diameter tubes achieved similar burst pressure values. Appendix A shows a tube after the burst test.

For tubes with a 6 mm outer diameter, a minor discrepancy of approximately 10% in burst pressure emerges, likely due to regions with lumps hindering material processing due to the elevated molar mass. It is worth noting that TPU containing 5% by weight of soybean polyol showcases excellent process stability throughout the extrusion process.

## 4. Conclusions

A series of high-molecular-weight thermoplastic polyurethanes (TPUs) consisting of fossil-based polyol and soybean-based polyol, MDI diisocyanate, and 1,4-butanediol extender have been successfully synthesized through reactive extrusion.

Using bio-based polyols with high functionality leads to branching and, above a certain content, to crosslinking. The segmental restriction promoted by cross-linking directly affected the glass transition values and the activation energy of the segmental mobility. This segmental restriction, in turn, had a significant impact on the phase separation and other properties of the TPU. Remarkably, the addition of 5 wt.% soybean polyol as a substitute for a fossil-based polyol resulted in satisfactory thermo-mechanical properties and processability, showing higher hydrophobicity. Hence, by employing polyols from renewable sources such as soybean oil, many physical–chemical properties of TPUs can be finely tuned, broadening their application range and allowing their use where fossil-based polyol TPUs present limitations.

Furthermore, increasing the soybean polyol content reduced both the number of hydrogen bonds and the surface tension. In addition, due to cross-linking, phase separation was also reduced with the use of bio-based polyol. Increasing the amount of bio-based polyol led to an increase in the T_gSS_ values, while the crystallinity degree tended to decrease. SAXS analysis revealed that the morphology of the systems was temperature-dependent, and due to formation and cross-linking, there was a limitation in mobility that could be activated by increasing the temperature. Notably, the domains in TPUs containing soybean polyol exhibited greater stability at high temperatures due to mobility restriction.

Materials generated through the utilization of renewable source-derived polyols, as demonstrated in this study, are suitable for applications demanding high molecular mass and processing stability. Nonetheless, it is advisable to employ them in quantities not exceeding 5 wt.%.

## Figures and Tables

**Figure 1 polymers-15-04010-f001:**
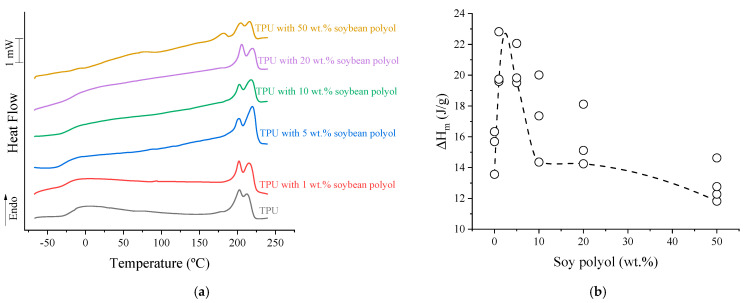
(**a**) DSC curves of TPUs for the second heating cycle; (**b**) enthalpy of fusion as a function of the soybean polyol content. The line refers to the spline, just to help with visualization.

**Figure 2 polymers-15-04010-f002:**
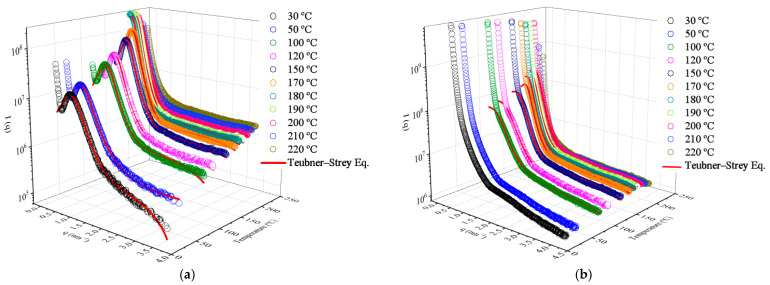
SAXS profiles: (**a**) TPU with 0 wt.% of soybean polyol at different temperatures; (**b**) TPU with 50 wt.% of soybean polyol. The continuous red line refers to the Teubner–Strey model fit.

**Figure 3 polymers-15-04010-f003:**
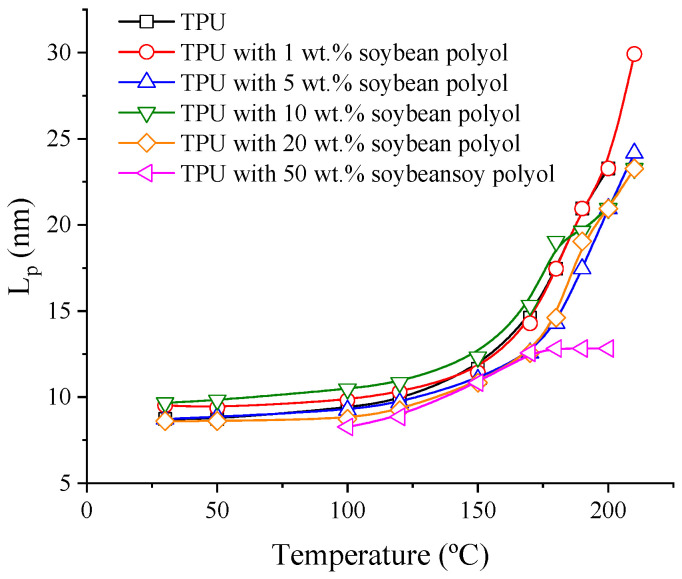
Long period (L_p_) obtained by Lorentz plot for TPU samples.

**Figure 4 polymers-15-04010-f004:**
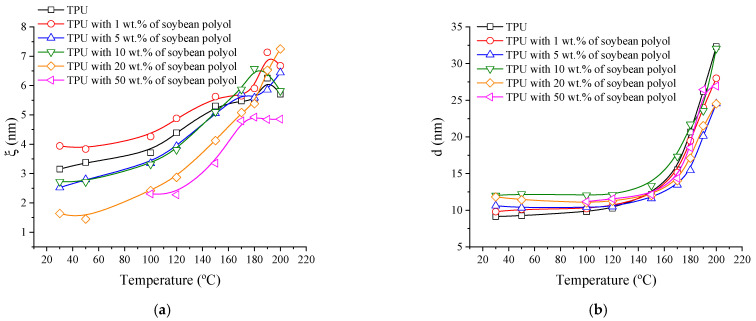
(**a**) Correlation length (ξ) and (**b**) mean repetition distances (d) vs. temperature.

**Figure 5 polymers-15-04010-f005:**
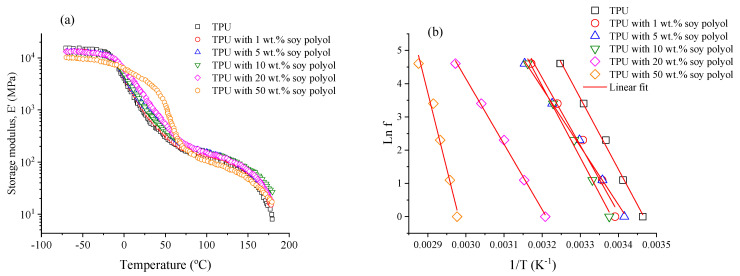
DMA results: (**a**) storage modulus (1 Hz) vs. temperatures for all TPU samples; (**b**) Arrhenius plot for the activation energy of segmental mobility (Ln f vs. 1/T).

**Figure 6 polymers-15-04010-f006:**
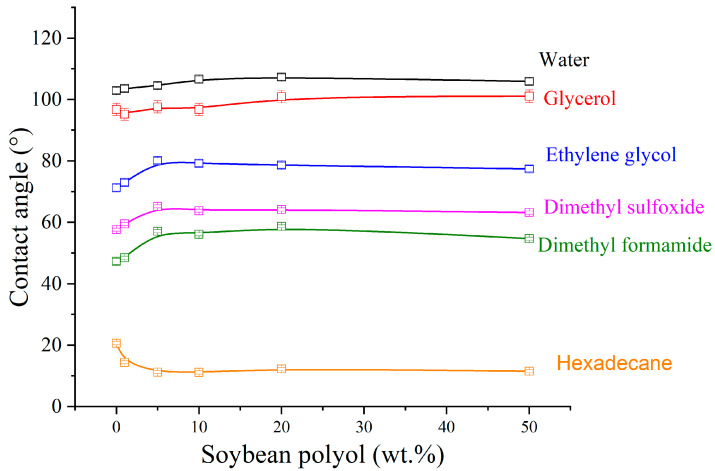
Contact angle vs. soybean polyol content.

**Table 1 polymers-15-04010-t001:** Molecular weight (M_n_, M_w_) and dispersity (*Đ*) values obtained by SEC.

Soybean Polyol Content (wt.%)	M_n_ (kDa)	M_w_ (kDa)	*Đ*
0	70.4	123.1	1.74
1	115.1	202.9	1.76
5	136.2	366.5	2.68
10 *	98.4	215.5	2.19
20 *	98.9	269.2	2.72
50 *	83.8	181.5	2.01

* Soluble fractions in DMF.

**Table 2 polymers-15-04010-t002:** Calculated parameters for determination of microphase separation: hard segment weight fraction in the polymer from initial molar ratios (f), weight fraction of hydrogen-bonded urethane groups (X_b_), weight fraction of hard segment-dispersed soft segments (W_h_), weight fraction of the mixed phase (MP), weight fraction of the soft phase (SP), and weight fraction of the hard phase (HP).

Soybean Polyol Content (wt.%)	f	X_b_	W_h_	MP	SP	HP
0	0.462	0.528	0.288	0.133	0.671	0.329
1	0.462	0.523	0.286	0.132	0.669	0.331
5	0.462	0.520	0.289	0.134	0.671	0.329
10	0.462	0.515	0.288	0.133	0.671	0.329
20	0.462	0.433	0.315	0.145	0.682	0.318
50	0.462	0.444	0.292	0.135	0.672	0.328

**Table 3 polymers-15-04010-t003:** T_gSP_, ΔC_pS_, (T_g_-T_gSS_), and phase separation for TPU with different amounts of soybean polyol.

Soybean Polyol (wt.%)	T_gSP_ (°C)	ΔC_pS_ [J/(g.°C)]	(T_gSP_-T_gSS_) (°C)	Degree of Phase Separation (DPS) (%)
0	−18.7	0.423	5.3	79.9
1	−22.6	0.403	1.9	76.3
5	−24.5	0.383	2.3	73.1
10	−17.2	0.377	12.4	72.8
20	−10.3	0.297	24.9	58.8
50	82.6	0.170	134.7	43.1

**Table 4 polymers-15-04010-t004:** T_g_ determined by DMA, storage modulus in the elastic plateau (E_e_′_0_), crosslink density (ν_e_), and activation energy of segmental mobility.

Soybean Polyol (wt.%)	T_g_ (DMA) (°C) *	E_e_′_0_ (MPa)	ν_e_ (mol.cm^−3^) × 10^−4^	E_a_ (kJ.mol^−1^)
0	15.5	158.4	1.040	177.8
1	21.7	178.7	1.043	168.9
5	19.6	205.6	1.050	145.9
10	23.0	194.4	1.078	179.1
20	38.5	195.3	1.074	163.6
50	62.3	98.9	1.103	415.5

* Determined at 1 Hz; E_e_′_0_ was determined by exponential fit according to Appendix A.

**Table 5 polymers-15-04010-t005:** Surface energy γ_L_ of the TPUs, including dispersive (γ_S_^D^) and polar (γ_S_^P^) contributions.

Soybean Polyol (wt.%)	γ_L_ (mN.m)	γ_S_^D^ (mN.m)	γ_S_^P^ (mN.m)
0	27.26	0.19	27.08
1	27.38	0.14	27.24
5	24.97	0.11	24.86
10	26.07	0.03	26.05
20	25.64	0.01	25.63
50	25.44	0.04	25.40

**Table 6 polymers-15-04010-t006:** Mechanical properties of TPU and tubes.

Properties	Reference TPU (without Soybean Polyol)	TPU with 5 wt.% Soybean Polyol
Hardness (shore D)	53 ± 2	52 ± 2
Tensile strength at 100%, MPa	11.6 ± 0.3	10.9 ± 0.2
Tensile strength at 300%, MPa	13.8 ± 0.3	15.3 ± 0.2
Tensile strength, MPa	27.8 ± 0.9	46.7 ± 3.1
Burst pressure, bar (6 mm × 1 mm)	33.7 ± 0.5	30.4 ± 0.9
Burst pressure, bar (8 mm × 1.25 mm)	30.9 ± 0.6	30.8 ± 0.9

## Data Availability

The raw data needed to reproduce these findings can be shared if requested from the authors.

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
