# Peer review of "Soybean-Based Polyol as a Substitute of Fossil-Based Polyol on the Synthesis of Thermoplastic Polyurethanes: The Effect of Its Content on Morphological and Physicochemical Properties"

_polymers, 2023, doi:10.3390/polym15194010_

Round 1

Reviewer 1 Report

Dear Authors,

Overall, it's an interesting manuscript and I think it's also quite well written. However, you should refine it a bit. Below are my comments point by point.

Line 73: Explain why gel structures will make recycling more difficult.

Line 76: Although you referred to the literature, describe the problem of producing these materials on a large scale.

Line 80: Hasn't such a study been done before? If not, write it down. If so, explain why it is so important. This will highlight the innovation of your research.

Then write the scientific purpose of your research. It must be specified correctly.

Line 96: All raw materials and research equipment used in the research should be described (name: manufacturer, city, country). Review the entire methodology.

Line 236: See comment on line 96.

Figure 6: There should be error whiskers on the graph. Also, is a linear scale ok in this case? In my opinion you should remove the line connecting the points.

Line 507: The mechanical properties are described quite briefly. In my opinion, the discussion of the results in this subsection is also poorly done. You should improve this. Whether the obtained results are close or far from the expected values.

If the results are similar, it may be worth performing elastic modulus tests in such a situation. Of course, you don't have to do that in this article.

Table 5: The table is missing a unit for mechanical strength. Generally, these values should be expressed in MPa.

Line 547: Please specify what perspectives the material you produce has. The conclusions above are rather statements.

Author Response

Reviewer 1:

Overall, it's an interesting manuscript and I think it's also quite well written. However, you should refine it a bit. Below are my comments point by point.

Line 73: Explain why gel structures will make recycling more difficult.

Response to reviewer: We appreciate your valuable feedback. We have made the necessary correction to the sentence. What we intended to convey is that the development of gelled structures poses challenges for recycling through conventional methods like extrusion and thermoplastic injection.

Line 76: Although you referred to the literature, describe the problem of producing these materials on a large scale.

Response to reviewer: We improved the sentence. There are several limitations on the use of these polyols, such as restrictions on companies adjusting their polymerization plans. Variability in the composition of batches of polyols and many others.

Line 80: Hasn't such a study been done before? If not, write it down. If so, explain why it is so important. This will highlight the innovation of your research.

Then write the scientific purpose of your research. It must be specified correctly.

Response to reviewer: We improved the sentence.

Then write the scientific purpose of your research. It must be specified correctly.

Line 96: All raw materials and research equipment used in the research should be described (name: manufacturer, city, country). Review the entire methodology.

Response to reviewer: We improved the methodology.

Line 236: See comment on line 96.

Response to reviewer: We improved the methodology.

Figure 6: There should be error whiskers on the graph. Also, is a linear scale ok in this case? In my opinion you should remove the line connecting the points.

Response to reviewer: We appreciate your valuable feedback. We have included the error bar in the graph. The line only shows the trend of the points

Line 507: The mechanical properties are described quite briefly. In my opinion, the discussion of the results in this subsection is also poorly done. You should improve this. Whether the obtained results are close or far from the expected values.

If the results are similar, it may be worth performing elastic modulus tests in such a situation. Of course, you don't have to do that in this article.

Response to reviewer: We value your constructive input, and we've made enhancements to the text. It's essential to note that, in the case of elastomers, measurements typically involve stress values at specific deformations due to their non-linear behavior. Including the Tensile values at 100% and 300% deformation is crucial for any prospective projects involving this material type. We'd like to clarify that in our study, we did not measure the elastic moduli.

Table 5: The table is missing a unit for mechanical strength. Generally, these values should be expressed in MPa.

Response to reviewer: We appreciate your valuable feedback. We corrected ourselves and included new data

Line 547: Please specify what perspectives the material you produce has. The conclusions above are rather statements.

Response to reviewer: We appreciate your valuable feedback. We improved the text.

Author Response

Reviewer 2

  1. Authors used soy-based polyol to prepare biobased polyurethanes. In the manuscript, they refer to soy-based polyol using different names: soy-based polyol, soy, or soybean polyol. I recommend you use the same nomenclature to refer to the polyol.

Response to reviewer: We appreciate your valuable feedback. We improved the text.

  1. In the introduction (Lines 54-59): -based polyols with biobased polyols from renewable sources has gained substantial attention, leading to the development of more sustainable TPUs. This simple approach potentially reduces the environmental impact of polyurethane production while providing unique physicochemical properties [1]. However, the effects of adding bio-based polyols on the Which are the effects of adding bio-based polyols on the final characteristics of TPUs that are partially understood? Could you mention them?

Response to reviewer: We appreciate your valuable feedback. We improved the text. We make it clear that these effects are in relation to the structure and properties.

  1. Line 260 and Table 1: (polydispersity, Mw/Mn). According to this, IUPAC has deprecated the use of the term polydispersity index, having replaced it with the term dispersity, represented by the symbol . Please, replace the name in the text and add the symbol in the last column of Table 1.

Response to reviewer: We appreciate your valuable feedback. We improved the term.

  1. In the discussion of the results obtained by FTIR (section 3.2.), the authors discuss the obtained values in Table 2. How do you determine the MP fraction (weight fraction of the mixed phase). There is an expected correlation between MP values and the values of the other fractions (SP and HP)?

Response to reviewer: We genuinely appreciate your valuable feedback. It appears that the reviewer may not have closely examined equations 1, 2, and 3. To clarify the procedure, it's essential to first establish the value of 'f,' representing the experimental fraction. Following that, we determine 'Xb' and 'Wh' while making necessary adjustments to the corresponding areas. Our research groups have a substantial body of work on this topic, providing valuable insights. Once these steps are completed, equation 3 can be readily applied. It's worth noting that the 'MP' fraction is a component within the 'SP' and 'HP' fractions.

Ernzen, J.R.; Romoaldo, C.H.; Gommes, C.; Covas, J.A.; Marcos-Fernández, A.; Fiorio, R.; Bianchi, O. Tuning Thermal, Morphological, and Physicochemical Properties of Thermoplastic Polyurethanes (TPUs) by the 1,4-Butanediol (BDO)/Dipropylene Glycol (DPG) Ratio. Polymers 2022, 14, 3164.

Favero, D.; Marcon, V.; Figueroa, C.A.; Gómez, C.M.; Cros, A.; Garro, N.; Sanchis, M.J.; Carsí, M.; Bianchi, O. Effect of chain extender on the morphology, thermal, viscoelastic, and dielectric behavior of soybean polyurethane. Journal of Applied Polymer Science 2021, 138, 50709, doi:https://doi.org/10.1002/app.50709.

  1. Lines 368-370: However, at when the temperature surpassed 170 °C, this power-law behavior shifted to a range of 0.5 < q < 1 nm-1

Response to reviewer: We appreciate your valuable feedback. We improved the text.

  1. Figure 2b: Please, add the red line corresponding to the Teuber-Strey Eq. in the legend of Figure 2b.

Response to reviewer: We appreciate your valuable feedback. We correct the figure.

  1. Line 386-389 The long period (Lp) of the samples was estimated as a function of temperature by plotting I(q).q2 vs. q (Lorentz correction), and the results are depicted in Figure 3. As the temperature increased, Lp also increased, indicating enhanced mobility and larger separation between the scattering centers. Observing the curves plotted in Figures S5, I have doubts about what are you plotting. Are you plotting I(q) vs. q.?

Response to reviewer: We appreciate your valuable feedback. I'm sending the data to the reviewer to make sure. How do you know when an I(q).q2 vs. curve is constructed? q is more accurate for determining Lp compared to I(q) vs. q. This is the basic treatment of SAXS curves.

Example of curve I(q) vs. q. for sample with 10wt. of soybean polyol

Example of curve I(q).q^2 vs. q. for sample with 10wt. of soybean polyol

  1. Lines 438-439 The elastic plateau region shifts to higher temperatures with increasing soybean polyol content due to the formation of cross-links. Observing the curves plotted in Figure S6-S11, it is true that the elastic plateau region shifts to higher temperatures. Furthermore, it becomes narrower since the minimum of the plateau moves to higher temperatures when the soybean polyol content increases and the maximum of the plateau keeps more or less constant (between 150 170 °C). Do you find an explanation for this trend?

Thank you for your insightful feedback. We value your input. In the context of forming partial cross-links, it is important to note that these connections give rise to distinct regions within the material, each characterized by varying barriers to conformational changes. As the extent of crosslinking increases, it gives rise to clusters of crosslinked material that tend to exhibit cooperative motion. Consequently, the heterogeneity in these cooperative movements leads to a broader plateau in the material's behavior. Conversely, as the population of heterogeneous cooperative movements diminishes, although the material may not be entirely reticulated, it results in a reduction in the plateau's extent.

Matsuoka, S. Mechanical relaxation processes in polymers. Applications to Polymers and Plastics 2002, 111-146.

Shen, J.; Lin, X.; Liu, J.; Li, X. Effects of Cross-Link Density and Distribution on Static and Dynamic Properties of Chemically Cross-Linked Polymers. Macromolecules 2019, 52, 121-134, doi:10.1021/acs.macromol.8b01389.

  1. Regarding the contact angle measurements, I recommend adding a figure with the image of the drop obtained (at least one with each solvent when 5 % of soybean polyol was added). You can incorporate this figure in the supporting information

Response to reviewer: We appreciate your valuable feedback. We add the figures.

  1. Table 4: Are the surface energy values, the dispersive and polar contributions average values? Or do you obtain them from the average of the contact angle measurements? If they are average results, please add the SD (Standard deviation).

Response to reviewer: We appreciate your valuable feedback. The values were obtained from the averages of the contact angles.

  1. Mechanical properties section: When you evaluate the mechanical properties of your materials, I recommend you add a figure with photos of the final tubes/samples prepared. You can incorporate this figure in the supporting information.

Response to reviewer: We appreciate your valuable feedback. We add the figures.

  1. Conclusions (Lines 535-538 Hence, by employing polyols from renewable sources, such as soybean oil, many physical-chemical properties of TPUs can be finely tuned, broadening their application range and allowing their use where fossil-based polyol TPUs present limitations. Which limitations present fossil-based polyol TPUs?

Response to reviewer: We appreciate your valuable feedback. We improved the text.

  1. Some figures of the supporting information are not cited in the main manuscript. Please, reference them in the text when you are discussing them.

Response to reviewer: We appreciate your valuable feedback. We improved the text.

  1. Figure S5: Be careful with the legends of the graphics included in this figure. The units of some temperature value are not complete e S5b and S5c)

Response to reviewer: We appreciate your valuable feedback. We improved the graphs.

All minor issues have been corrected in the text

Round 2

Reviewer 1 Report

I accept the corrections made.

Author Response

We thank the reviewer for his work and for accepting the manuscript with the corrections carried out under his/her advice

Reviewer 2 Report

I attached all my suggestions in the field: “Reviewer - Comments for authors”.

Author Response

We thank the reviewer for his/her work.

We have made all the corrections proposed by the reviewer, including the addition of a space between number and units throughtout the manuscript (text and figures 1 to 5).

Respect to the meaning of the second "s" in Tgss, the maning of "ss" is "soft segment", as stated in the Introduction, line 43.